# The varying impacts of COVID-19 and its related measures in the UK: A year in review

**Muzhi Zhou** *, Man-Yee Kan

Department of Sociology, University of Oxford, Oxford, United Kingdom

* muzhi.zhou@sociology.ox.ac.uk

## Abstract

We examine how the earnings, time use, and subjective wellbeing of different social groups changed at different stages/waves of the pandemic in the United Kingdom (UK). We analyze longitudinal data from the latest UK Household Longitudinal Survey (UKHLS) COVID study and the earlier waves of the UKHLS to investigate within-individual changes in labor income, paid work time, housework time, childcare time, and distress level during the three lockdown periods and the easing period between them (from April 2020 to late March 2021). We find that as the pandemic developed, COVID-19 and its related lockdown measures in the UK had unequal and varying impacts on people's income, time use, and subjective well-being based on their gender, ethnicity, and educational level. In conclusion, the extent of the impacts of COVID-19 and COVID-induced measures as well as the speed at which these impacts developed, varied across social groups with different types of vulnerabilities.

## Introduction

More than one year has passed since the United Kingdom (UK) officially announced its first national lockdown on 23 March 2020 due to the rapid spread of COVID-19. The outbreak of COVID-19 and the massive lockdown measures have greatly changed people's lives. When people were instructed to stay at home and maintain physical distancing, the lives of millions of people were affected. For months, many people were unable to go to work or school, nor could they meet friends and relatives. What was unexpected was that people in the UK experienced a total of three national lockdowns over the past year. Now, people's lives are far from what they were before the first lockdown, and the pandemic is still not over.

Recent evidence has shown that the COVID-19 pandemic and related social and economic measures, such as physical distancing and business closure, have differential impacts on various social groups. In the UK, for example, women and parents are found to have experienced a larger reduction in subjective wellbeing [1, 2]. Black, Asian, and minority ethnic (BAME) immigrants were more likely to experience economic hardship immediately after the first national lockdown [3]. In addition, among those who were known to have COVID-19, people of BAME background in the UK had a death rate that was higher than that of white people [4]. As Damian Barr said in his poem, "we are in the same storm, but we are not all in the same boat [5]".

id=8644 https://beta.ukdataservice.ac.uk/
datacatalogue/studies/study?id=6641.

**Funding:** This work is supported by the European
Research Council (ERC) under the European
Union's Horizon 2020 research and innovation
programme (awardee: Man-Yee Kan, grant number
771736). Funding website: https://ec.europa.eu/
programmes/horizon2020/en. The funders had no
role in study design, data collection and analysis,
decision to publish, or preparation of the
manuscript.

**Competing interests:** The authors have declared
that no competing interests exist.

These earlier findings identified the existence of immediate unequal impacts for different social groups, but our understanding of the longer-term impacts of COVID-19 and related measures remains limited. We know little about how the impacts might have changed since the first lockdown. The COVID-19 pandemic has already lasted for more than one year, and the UK has experienced three national lockdowns. Early research was confined by data that covered only two time points—such as before and shortly after the announcement of the first lockdown. Little is known about to how unequal social impacts reveal themselves at different stages of the COVID-19 pandemic, especially with repeated lockdowns. This omission hinders our understanding of how COVID-19 and COVID-induced social policies, such as physical distancing measures, working from home, and the closure of certain businesses, which have been changing on a weekly or even daily basis, progressively affect people's lives. Documenting the development of the impacts of COVID-19 and COVID-induced measures is important for us to understand the consequences of this rapidly developing pandemic and help policymakers plan for future waves and future pandemics.

We need more comprehensive and up-to-date research on how inequalities have changed as the COVID-19 pandemic develops with repeated waves and the various measures to contain it were implemented over the past year. We conducted analyses on a nationally representative population data from the latest UK Household Longitudinal Survey (UKHLS), which was conducted before the first lockdown in March 2020, during the first lockdown from April to June 2020, during the ease of the first lockdown (June to September 2020), and during the later two lockdowns (November 2020, and from January 2021 to March 2021). In this paper, we contribute to COVID-19 research by providing a dynamic picture of how people's labor earnings, time use, and wellbeing changed across different stages of the pandemic. We further investigated whether and the extent to which the inequalities in these outcomes based on gender, ethnicity, and educational level have changed over the past year.

In what follows, we first review the latest works on the impact of COVID-19 and COVID-induced measures on people's lives, focusing on three dimensions of social inequality: gender, race/ethnicity, and education. We then outline the development of the COVID-19 pandemic and the lockdown measures in the UK from March 2020 to April 2021. Next, we introduce the data and its longitudinal design, which enables us to compare the information of the same individuals before the start of this pandemic and at different time points over the past year. Finally, we will report the results of fixed-effect regression analyses and discuss our conclusions.

## The impacts of COVID-19 and its related measures

The COVID-19 pandemic has developed for over one year. In many countries, repeated waves of COVID-19 have been observed. The primary aim of COVID-19 induced measures is to contain the virus by reducing physical contacts between people. Many of these measures immediately affect people's behaviors, but others could have longer-term impacts. For example, the closure of businesses and work-from-home guidance tremendously altered people's working patterns. Reductions in paid work time and earnings have been immediately recorded in countries that have introduced lockdown measures such as Australia [6], the UK [3, 7], and the United States (US) [8]. When more people stayed at home and the option of outsourcing domestic work was reduced due to business closure or the fear of contracting COVID-19, it is not surprising to see that people spent substantially more time on unpaid domestic work than they had in the past [6, 7, 9, 10].

People's feelings also changed. The contraction of COVID-19 is associated with a series of symptoms such as a high temperature, continuous cough and a loss or change to the sense of

smell or taste. Serious cases will result in hospital admission and death. In the UK, the case-fatality rate is estimated to be 2.1% [11]. Daily news reporting the surging number of new cases and deaths brings in a high level of worry about health and security [2]. In addition, loss of employment, financial strain, and social isolation are well-known factors that negatively affect mental health [12–14]. Not surprisingly, soon after the start of the pandemic, worsened subjective wellbeing was observed in Australia [6, 15], the UK [2, 16, 17], and the US [18]. Once daily increase of COVID-19 cases declined and the lockdown restrictions began to be lifted, people's subjective wellbeing started to recover. As Pierce et al. [2] noted by using the first five waves of the same UKHLS COVID study data as in this paper, "[b]etween April and October 2020, the mental health of most UK adults remained resilient or returned to pre-pandemic levels." However, "[a]round one in nine individuals had deteriorating or consistently poor mental health."

This COVID-19 pandemic and its related measures have raised increasing concerns of exacerbated social inequalities. Since long before the pandemic, gender inequalities have existed in the labor market. In the UK, the labor force participation rate for men is higher than that for women, and men are also much more likely to work full time [9, 19]. Women are more likely to be at-home workers. Reasons for this inequality include inflexible workplace expectations, gender norms expecting men to be the primary earners and women the primary caregivers, and discrimination in the labor market. When people are required to work from home, the spatial boundary between market work and family life is blurred. Many studies have investigated whether the changes in time use due to lockdown measures are the same for women and men. Between March and May 2020 (UK 1st lockdown), British men were found to be more likely to be furloughed or dismissed from work than women [20]. However, studies focusing on the labor market performance of parents reveal a different pattern. In the UK, during the first lockdown period from April to May 2020, among parents with children aged between 4 and 15, mothers were found to be more likely to be laid off, furloughed, or quit their jobs [21]. Similarly, in Australia [6], Canada [22], and the US [23], mothers with young children experienced a larger change in their paid work time or were more likely to leave their jobs. On the other hand, several studies have reported improvements in the domestic division of labor: the increase in domestic work was larger for men than for women during the lockdown period in Australia [6], Canada [24], France [25], and the US [26]. However, contrary results were reported in Germany [27] and Spain [28]. The decline in subjective wellbeing also differs between women and men. In the UK and Australia, women were found to experience a larger reduction in subjective wellbeing than men [1, 2, 6, 9, 29].

In the UK, BAME immigrants were more likely to experience economic hardship just after the first lockdown [3]. Compared with their white counterparts, BAME immigrants were also found to suffer a larger decline in subjective wellbeing at the beginning of the March 2020 lockdown in the UK [3, 30]. In the US state of Indiana, Black Americans were more than three times more likely to lose their jobs than whites [31]. In contrast, another study highlights that white Britons in middle-income jobs were more likely to experience job loss, primarily driven by the fact that many BAME people are employed in key sectors such as the health and social care services, which were exempt from the lockdown measures and instead had a surge in work demands, during the first UK lockdown [20]. Notably, in the UK, people of BAME backgrounds had a death rate that was higher than that of white people after they were confirmed to have COVID-19 [4].

People with less education and lower income suffered substantially during the pandemic. They were particularly hit hard with a higher chance of losing their jobs and earnings in countries such as Canada [32], the UK [20], and the US [31]. Many of the less educated are trapped in lower-skilled occupations with tight financial constraints. Consequently, the less educated

group reported a heightened level of distress during the first lockdown in the UK [33]. However, one US study reports that the decline in subjective wellbeing up to April 2020 was larger among the more educated, possibly because the more educated might have felt a greater loss of control and wealth due to COVID-19-related uncertainties [18]. Another study conducted in the US between April 2020 and June 2021 pointed out that part of the reason for the deterioration of mental health results should be attributed to the concurrent presidential election and unrest in domestic politics [34].

Again, the current literature has focused extensively on the impacts of the relatively early stage of this pandemic. In particular, studies that have employed the same British data source as the present study have examined the changes in earnings, time use, and subjective wellbeing during the implementation of the first national lockdown in late March 2020 [3, 7, 9, 10, 20]. Pierce et al.'s work [2] on subjective wellbeing is an exception. Their work examined the recovery of subjective wellbeing when the first lockdown measures were eased from June to October 2020. However, their study did not cover the later lockdowns in November 2020 and January 2021. In this article, we will provide a first-year review of COVID-19 development in the UK and document how people have responded to the first lockdown, the ease of the first lockdown, and the later two lockdowns. This evaluation will reveal whether people responded similarly to repeated lockdowns and whether these changes in earnings, time use, and feelings are temporary or long-lasting.

## Timeline of the lockdown measures in the UK

On 31 January 2020, the first two positive cases of COVID-19 were confirmed in the UK. On 5 March 2020, the first patient who tested positive for COVID-19 died. On 23 March 2020, the Prime Minister placed the UK on lockdown to slow down the outbreak of this pandemic. These measures included physical distancing, school closures, working from home, and closure of non-essential businesses, including pubs and cafes. Key sectors, including health and social care, education and childcare, and key public services, were allowed to operate.

To maintain employment and to protect individuals and businesses from economic hardship, a coronavirus job retention scheme was implemented for the period between late March and the end of October 2021 to cover 80 percent of the regular salary of furloughed employees, up to a maximum of £2,500 per month [35]. In April, the UK had more than 10,000 deaths related to COVID-19. In May, phased reopening of shops and schools was announced, and those who were unable to work from home were expected to return to the workplace.

Beginning on 1 June 2020, schools were open for all Reception, Year 1 and Year 6 pupils, but the summer holiday soon arrived. Nonessential businesses reopened gradually beginning on 15 June. Beginning on 4 July, pubs, cinemas, restaurants reopened. Physical distancing rules were relaxed from a "two-meter" to a "one-meter plus" rule. In August, restrictions were eased further, although the pandemic was far from over.

The UK variant of the coronavirus (scientific name B.1.1.7, WHO name Alpha) was first identified in September 2020 and was considered to be more transmissible and potentially deadlier. In late September, people were required to work from home with a 10 pm curfew for the hospitality sector. In October, England entered a 3-tier system where different regions were classified into different tiers depending on the level of the spread of the virus. Soon after, the second national lockdown came into force on 5 November and lasted until 2 December. People were told to stay at home. Other measures included the closure of the hospitality sector and nonessential shops, but schools were open, and people could leave their home for outdoor exercise. After 2 December, the UK then entered a stricter 3-tier restriction system.

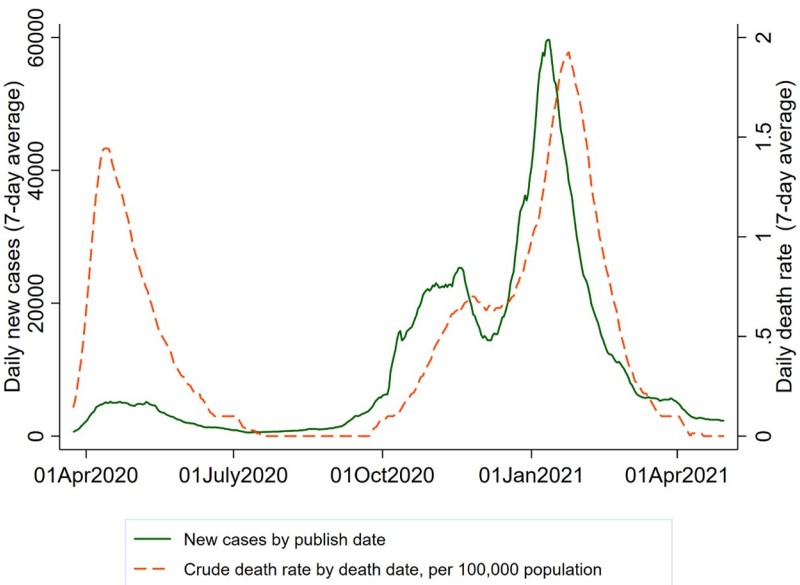

**Fig 1. COVID-19 pandemic in the UK.** Note: Data source: https://coronavirus.data.gov.uk/details. Crude death rate is new deaths within 28 days of a positive test per 100,000 population.

However, this 3-tier system did not last long. After Scotland announced a lockdown, on 4 January 2021, a third national lockdown was announced. Schools were closed again, and people were urged to stay at home. This time, the measures were stricter than those in the second lockdown. They included "Stay at home at all times, wherever possible," "Not allowed to meet others from outside your household (or support bubble)," "All retail and hospitality venues must close," and "Personal care services have to close." Schools were closed to most pupils, except for the children of critical workers and the most vulnerable children. Nurseries were kept open.

Since 8 March, schools in the UK have been completely reopened. Nonessential retail and personal care services have been reopened since 12 April. People have been allowed to meet outdoors, as a number of restrictive measures have been lifted since 17 May. A complete easing will occur on 19 July 2021. The Prime Minister has pledged that all adults in the UK will be offered their first dose of a COVID-19 vaccine by the end of July.

By 16 April 2021, the recorded number of deaths related to COVID-19 had reached over 127,000 in the UK. Fig 1 displays the spread of COVID-19 and related deaths in the UK during the research period. A more detailed timeline of the UK lockdowns can be found at [https://www.instituteforgovernment.org.uk/sites/default/files/timeline-lockdown-web.pdf]. Fig 1 shows the development of the COVID-19 pandemic in the UK based on data provided by the UK government.

## Data and methods

### Data and sample

We use data from the first eight waves of the UKHLS COVID study data and the preceding two waves (2017/18 and 2018/19) of the UKHLS main survey [36]. The UKHLS is a household panel survey and started its first wave in 2009 with a nationally representative sample of 51,000 adults (aged 16 and above) from approximately 40,000 households. Individuals were followed up annually and were interviewed face-to-face. This research is based completely on the

UKHLS data that are publicly available through the UK Data Service (Study numbers: 6614 and 8644) and are completely anonymous.

Regarding the COVID study, households who participated in previous UKHLS surveys were contacted to fill in a monthly online questionnaire beginning in April 2020. The complementary telephone survey started in May 2020. Participation in the survey was voluntary. Approximately 16,000 respondents (aged 16 and above) completed this first wave of the COVID survey with a response rate of 42%. Currently, data from the first eight waves of surveys conducted in the last week in April, May, June, July, September, November in 2020 and the last week in January and March in 2021 are available.

Our analytic sample contains individuals who have participated in the UKHLS main survey and at least one of the eight waves of the COVID study. The respondents all had access to the internet or telephone to participate in the surveys. This requirement might have caused a sample selection bias. In a supplementary analysis, the sample from the COVID study is found to be socioeconomically advantaged in terms of employment, occupation, education, and home-ownership compared to the full UKHLS sample. If we assume that one's socioeconomic status has a protective effect on the negative consequences of the COVID-19 and related lockdown measures, the reported results may underestimate the potential negative impacts of the COVID-19 and the related lockdown. Nonetheless, one paper discusses this issue of nonrandom sample selection and demonstrates that the bias due to sample selection is very limited once weight is considered [37]. In the following analysis, we apply the individual weights, which were adjusted for "unequal selection probabilities and differential nonresponse" and are supplied in the data [38]. Based on the User Guide for the data, these weights "scale respondents to the eligible population in the UKHLS wave 9 sample, adjusted for death, incapacity and emigration occurring between wave 9 and the start of the COVID-19 web survey." [38] This approach has been used in previous work analyzing the same data [2, 3, 20].

Our sample includes respondents of prime working age (between 20 and 65) in 2020. Two percent of the UKHLS COVID sample has missing values in the predictors to be used in regressions. The numbers of observations with no missing predictors are 10484, 9008, 8478, 8210, 7642, 7083, 7019, and 7525 in the first eight waves of the COVID study. The final sample for each regression is dependent on the outcome variables with nonmissing values (some outcome variables are not asked in certain waves) and the selection of subgroups (for example, people who had a job before the pandemic). Please refer to S1 Table for more details of the sample selection process. The focus on within-individual changes in the outcome variables indicates that the respondents should be followed up for more than one wave. Previous analyses using the same data and selecting the individuals interviewed for more than one wave do not find that this selection would bias the results [39].

## Measures

Monthly labor income, weekly paid work hours, subjective wellbeing, weekly housework hours, and weekly childcare hours are the five dependent variables or outcomes of interest.

**Monthly labor income.** Respondents' labor income in January or February 2020 (before the lockdown) was collected retrospectively in the COVID survey. Respondents also provided their current labor income in each month thereafter. We calculate the natural log of the labor income. Those who had a job in January or February 2020 were selected to predict this outcome.

**Weekly paid work hours.** Respondents retrospectively reported their current paid work hours per week and their usual working hours in January or February 2020. During the period of the COVID-19 pandemic, the question asked was "How many hours did you work, as an employee or self-employed, last week?" During the prepandemic period, the question was

"During January and February 2020, how many hours did you usually work per week?" Those who had a job in January or February 2020 were selected to predict this outcome.

**Subjective wellbeing.**   Subjective wellbeing is the mental wellbeing reported by the respondents in a General Health Questionnaire (GHQ). The value is the sum of 12 items (GHQ-12) scored on a Likert scale from 0 to 3: "ability to concentrate," "losing sleep," "playing a useful role in life," "capability of making decisions," "feeling under stress," "overcoming difficulties," "ability to enjoy activities," "ability to face problems," "feeling unhappy or depressed," "losing confidence," "believing in self-worth," and "feeling generally happy." The overall scale ranges from 0 (least distressed) to 36 (most distressed). This measurement is a validated and widely used measure of nonspecific mental distress in surveys [40]. The same information was collected in earlier waves of the main survey of the UKHLS and in each wave of the COVID study. The full sample was used to predict this outcome.

**Weekly housework hours.**   Respondents' weekly housework hours were collected by the question "Thinking about last week, how much time did you spend on housework, such as time spent cooking, cleaning and doing the laundry?" Information about housework hours before the COVID survey was derived from the earlier UKHLS waves (the latest one was collected in the years between 2018 and 2019). The full sample was used to predict this outcome.

**Weekly childcare hours.**   Respondents' childcare hours were collected by the question "About how many hours did you spend on childcare or home-schooling last week?" This information is only available in the COVID survey. Only those who had a child younger than 16 years old in the household (referred to as parents in later analyses) were asked this question, and these respondents are used for analyses.

**Independent variables.**   We include the wave dummies, which represent the time point when information was collected to examine the dynamics in those outcome variables.

The key socioeconomic independent variables are constant for the same individual across the waves. These variables are gender (52.7% females), whether an individual is Black, Asian or another minority ethnic (10.1%) or not (reference group: whites), and educational level (university degree holders 32.2%). The underrepresentation of ethnic minority groups is common in a panel survey sample (the 2011 census reported that 85.6% of the working-age people were from white ethnic groups) because of the selection of people with repeated observations to satisfy the requirement of the fixed-effect models. People with disadvantaged backgrounds are known to be more likely to drop out in repeated surveys [41]. The later regression analysis has considered this sample selection issue using weights, as discussed above. Moreover, attrition in panel surveys is not found to have a significant impact on the estimations in predicting income [42], time use [43], or attitudes [44].

Whether the respondent had a positive COVID-19 test outcome was asked in each wave. We included this variable in the model to control for the impact of contracting COVID-19 so that the period indicators could better represent the spread of COVID-19 and COVID-19-related policy change at the macro-level. This variable has four categories: "having no test" (reference, 89.7%), "tested positive" (0.8%), "tested negative" (9.0%), and "result pending" (0.5%).

All models controlled for respondents' partnership status (whether they live with a partner) and parenthood status (the presence of a child younger than age 16 in the household) to account for potential changes in the family status that are correlated with the outcomes [45, 46].

## Analytical strategies

We applied linear fixed-effect regressions to predict the five outcomes. By interacting the month indicator with gender, BAME group, and education levels, we examined how the

change in income, time use, and wellbeing differed across individuals in the three different sociodemographic groups in different periods of the pandemic. The reference time point is January and February 2020 for earnings and weekly paid work hours outcomes. The reference time point is the year 2018/2019 for the subjective wellbeing (distress level) and weekly housework hours outcomes. For weekly childcare hours, the reference time point is April 2020, which was during the first national lockdown. The outcome variables compare the information reported by the same individuals at each time point and hence reveal within-person changes. This analytic approach enabled us to investigate trajectories of the outcome variables over the past year conditional on the same individual.

The fixed-effect regression method takes full account of the time-constant individual characteristics that are correlated with both the independent variable and the outcome variables. This is achieved by demeaning the dependent and independent variables using person-specific means [47].

The samples in the UKHLS main survey and the COVID survey are probability samples of postal addresses. The samples are clustered and stratified. Accordingly, clustered standard errors are used to consider this sampling design [48].

These analyses were conducted in Stata/SE 16.1. Replication codes are available at https://github.com/jomuzhi/ukcovidunderstandingsociety.

## Results

### Descriptive results

We first report the weighted mean values of the key outcomes in Table 1. Please note that the information was collected at the end of each survey month.

First, among those who worked before this pandemic (between January and February 2020), there was a clear reduction in their average earnings when the pandemic started in the UK. Their income recovered by almost ten percent in May from the April level, which should have been mainly driven by the implementation of the *job retention scheme*. Some workers who could not work from home, such as those working on construction sites, also returned to the workplace in May. Since then, average monthly net earnings have remained at approximately the level of £1,550. Notably, since the first lockdown, people's take-home earnings has never returned to their prepandemic level but never fell below 90% of the pre-pandemic level.

Table 1. Mean values of earnings, paid work hours, subjective wellbeing, housework hours and childcare hours in each month.

| COVID-19 lockdown stage | Date | Net earnings [Mean] | Weekly working hours [Mean] | Distress level [Mean] | Weekly housework hours [Mean] | Weekly childcare hours [Mean] |
|---|---|---|---|---|---|---|
| Pre-COVID | Jan/Feb 2020 or 2017/19 | 1,673.9 | 34.7 | 11.8 | 9.0 | - |
| 1st lockdown | Apr 2020 | 1,428.3 | 21.9 | 13.1 | 12.3 | 16.7 |
| 1st lockdown | May 2020 | 1,570.2 | 23.9 | 13.1 | 12.2 | 15.8 |
| School reopened | Jun 2020 | 1,546.5 | 26.0 | 13.1 | 11.3 | 13.3 |
| Easing | Jul 2020 | 1,559.7 | 27.1 | 12.4 | - | - |
| Easing | Sep 2020 | 1,553.2 | 30.2 | 12.4 | 10.4 | 12.9 |
| 2nd lockdown | Nov 2020 | 1,573.1 | 29.0 | 13.4 | - | - |
| 3rd lockdown | Jan 2021 | 1,561.0 | 28.1 | 13.4 | 10.5 | 13.4 |
| Schools reopened | Mar 2021 | 1,547.6 | 28.6 | 12.9 | - | - |

Note: Cross-sectional baseline respondent weight was applied to represent the population of all adults (16+) who were resident in private households in the UK at the time of wave 9 (2017/18), and who did not die or emigrate before the relevant web survey. Data source: UKHLS main survey waves 1 to 10 (up to 2019) and COVID study waves 1 to 8.

Before the pandemic, those who worked in January and February 2020 worked 34.7 hours per week on average. A record low of 21.9 hours per week was observed in April 2020. The persistent decline in paid work time over the past year is evident, although working hours have recovered gradually since May and reached a peak of approximately 30 hours per week in September 2020. The later two national lockdowns (November 2020 and January 2021) did not reduce the working hours as much as the first national lockdown. Weekly paid work hours were maintained at approximately 28 hours.

People felt more distressed beginning in March 2020. The worst number of 13.4 was recorded in the last two rounds of lockdown-November 2020 and January 2021, when new cases and deaths grew sharply at the beginning of these lockdowns.

People's housework hours increased and reached the highest level of 12.3 hours per week in April and May 2020. Then, housework time declined gradually and was maintained at 10.5 hours per week. Compared with the figure recorded in September 2020 when most lockdown restrictions were eased, the figure in January 2021 did not change significantly, even though a stricter lockdown was in place. This finding concurs with the small reduction in paid work hours from September 2020 to January 2021.

The average childcare hours per week reached 16.7 hours for parents in April, but this figure gradually declined to approximately 13 hours per week before the third national lockdown. In January 2021, childcare hours only increased 0.5 hours per week over the September figure, even though schools were closed to most pupils during the third lockdown. Overall, people's time use had become less responsive to repeated lockdowns.

## Changes in earnings, paid work time, subjective wellbeing, housework and childcare time

Fig 2 reports within-individual changes in earnings, paid work hours, distress level, and housework hours across waves. The red lines indicate the time point when the national lockdowns

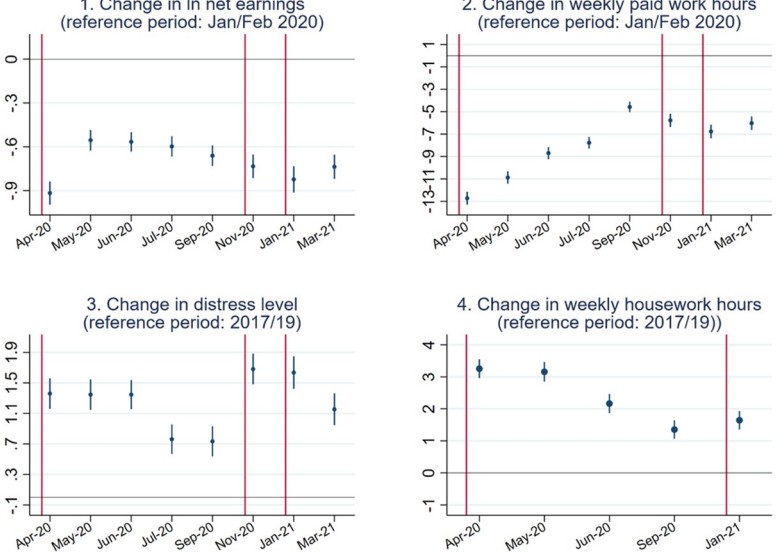

**Fig 2. Point estimates and 95% confidence interval of the baseline models.**

started to enforce. Please note that the information was collected at the end of each survey month. Detailed coefficients are reported in S2 Table.

Respondents' earnings stayed lower than the pre-pandemic level over the entire year, with the largest decline (~9%) recorded in late April, the first month after the announcement of the first national lockdown. Earnings recovered slightly after the gradual relaxation of restrictive measures and the implementation of the job retention scheme. Following the third lockdown, when almost the same strict measures as the first lockdown were imposed, we found a similar level of decline in earnings (~8%) compared with the prepandemic period, as in the first lockdown. One year after the onset of the pandemic in the UK, our sample still experienced a 7.4% decline in earnings compared with the pre-pandemic level.

Paid work hours remained much lower than the prepandemic level over the entire year. The largest drop of nearly 13 hours was observed in the first month after the March 2020 lockdown. Then, paid work hours recovered and have never returned to the same lowest point. People worked the longest hours in September 2020, when restrictive measures were minimal. Interestingly, despite the implementation of the second and the stricter third national lockdowns, paid work hours dropped only slightly compared to the September figure and were even higher than the July 2020 figure, even though all shops were allowed to open back in July 2020. This observation suggests an increased adaptation to the work-from-home practice. After the first lockdown, more firms announced a long-term strategy to allow employees to work from home [49]. Accordingly, people have increased their paid work time even though they might still work from home.

In this pandemic, people's subjective well-being has been damaged. The distress level (a higher score indicating more distress) stayed higher than the prepandemic level over the past year. In the three-month period after the first lockdown, a high level of distress was recorded. An improvement in subjective wellbeing was observed from July and before the enforcement of the second lockdown. The November lockdown brought a further decline in subjective well-being, which is consistent with the findings in one earlier study [2]. The distress level in November 2020 and January 2021 was even higher than that in the first lockdown period. It appears that people were much less optimistic and suffered tremendously as the pandemic dragged longer. People became slightly less negatively affected in their subjective wellbeing in March 2021, although the level was only similar to that in April 2020. One year after the onset of the pandemic in the UK, respondents' subjective wellbeing returned to the level of April 2020, which was one month after the announcement of the first national lockdown.

The increase in housework hours was the highest during the first lockdown. Compared with the housework hours during the easing period in September 2020, the January 2021 lockdown was not associated with an increase in people's housework time. This change echoes the relatively high level of paid work time in the later two lockdown periods.

Next, we examine childcare time since the first national lockdown. In Fig 3, we can see that beginning in April 2020 (during the first lockdown period), childcare hours have been dropping. The lowest level was observed in September 2020, when schools completely reopened. Interestingly, childcare hours in January 2021 were similar to those in September 2020, despite the closure of schools to most children in January 2021.

## Differential impacts on women and men

Figs 4 and 5 report whether changes in the five indicators differ between women and men. For monthly net earnings and weekly paid work hours, we analyzed an additional sample that includes only non-key workers. We will examine whether a disproportionate number of female workers in certain key sectors, such as health and social care, drive the results.

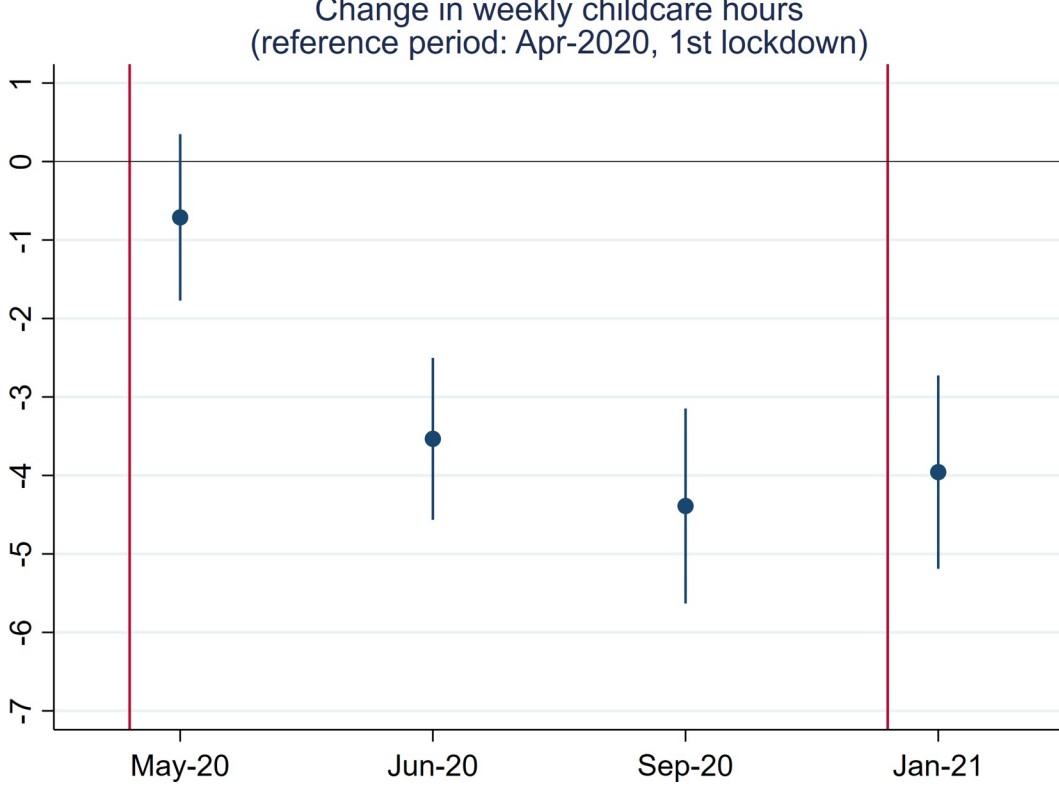

**Fig 3. Point estimates and 95% confidence interval of the baseline model to predict childcare time.**

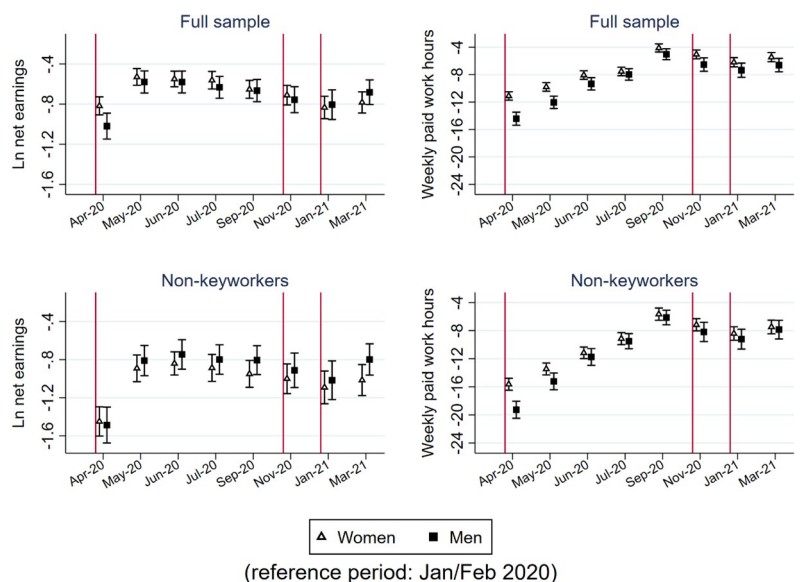

**Fig 4. Point estimates and 95% confidence interval of the gender-period interaction models to predict earnings and working time.**

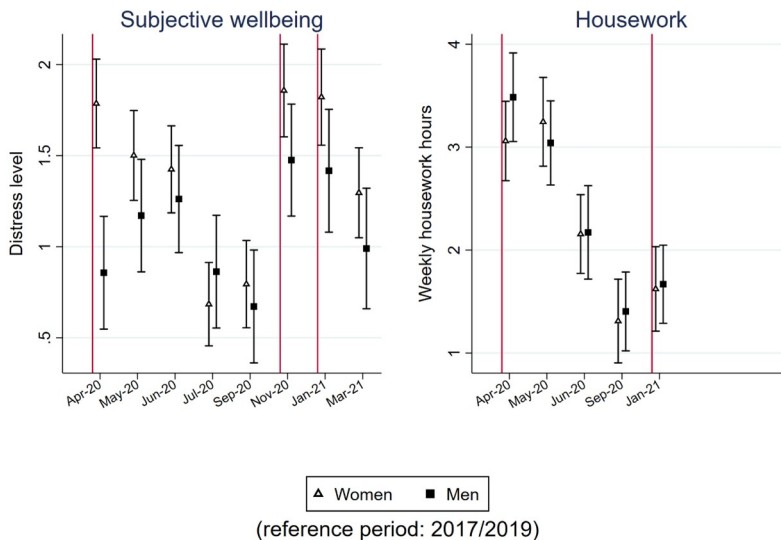

**Fig 5. Point estimates and 95% confidence interval of the gender-period interaction models to predict subjective wellbeing and housework time.**

First, the reduction in earnings for female workers (those who worked in Jan/Feb 2020) was smaller than that for male workers during the first lockdown in April 2020 (p = 0.011). Since then, there has been no difference between women and men in changes in earnings, reflecting the faster recovery of men's earnings. Differential impacts on women and men were not found among non-keyworkers. Therefore, the higher proportion of women working in key sectors, which were operating much more actively than other sectors during the first lockdown period, should be the main reason for the gender difference in the earning decline during the first lockdown.

During the first lockdown, the decline in paid work hours was smaller for female workers than for male workers, disregarding their keyworker status (p<0.001). The gender difference in the reduction in paid work hours decreased as the first lockdown ended and became statistically insignificant at the 0.05 level from July to September 2020, indicating a faster recovery of paid work time for men than for women. The differential impacts of gender on paid work hours observed in the first lockdown were not observed in later lockdowns among non-keyworkers.

In Fig 5, the growth in distress level was much higher for women than for men in the first month of the first lockdown (p<0.001). Then, women's subjective wellbeing recovered, and men's distress levels began to rise. These findings suggest that men's response to this pandemic lagged behind that of women in terms of their subjective wellbeing in the first lockdown. The distress level of both women and men was reduced to the lowest level from July to September 2020, when life in general had returned to normal. Once the cases of COVID-19 surged and lockdown restrictions were reimposed in November 2020 (p = 0.056) and January 2021 (p = 0.061), women again suffered from a larger increase in distress levels than men. The distress level of women reached a similar high point across the three lockdowns. For men, their distress level was higher in the later lockdowns than in the first lockdown, when the cases of COVID-19 and its related deaths worsened.

We do not observe a gender-specific impact on housework time. The gender gap in housework time was maintained over the past year.

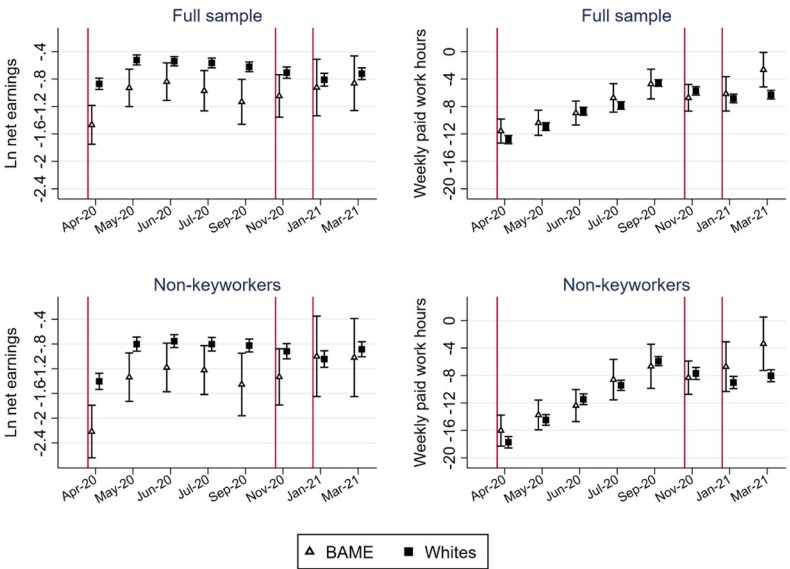

**Fig 6. Point estimates and 95% confidence interval of the race-period interaction models.**

## Differential impacts on BAME people and white people

Figs 6 and 7 report whether changes in the five indicators differ between BAME people and whites. For monthly net earnings and weekly paid work hours, we analyzed an additional sample that includes only non-key workers.

Compared with whites, the earnings of the BAME group were particularly negatively affected by the pandemic. The differential impacts on earnings persisted across almost all months over the past year, except during the third lockdown. The gap was large even when most lockdown restrictions were eased in September 2020 (p = 0.003). The earning gap between the BAME group and whites was even larger among non-key workers. Over the past

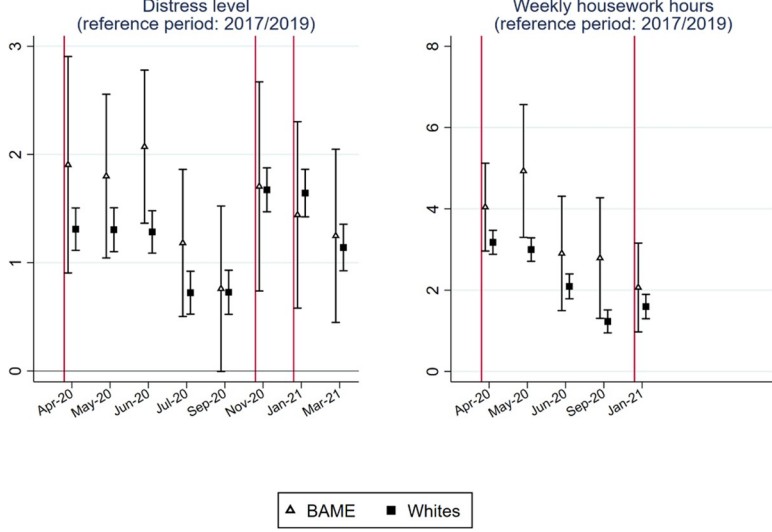

**Fig 7. Point estimates and 95% confidence interval of the race-period interaction models to predict subjective wellbeing and housework time.**

year, the decline in market working time was similar for the BAME group and whites in both the full and the non-key worker samples. In March 2021, the reduction in paid work time decreased less for the BAME group than for the whites (p = 0.006).

Regarding the distress level (Fig 7), the increase for the BAME group was larger than that for whites during the first lockdown, but the difference was not statistically significant at the 0.05 level due to the large standard error of the estimates of the BAME group. Beginning in September 2020, the changes in the distress levels were similar for the BAME group and whites. The increase in housework hours seems to be larger for the BAME group, but the large standard errors prevent us from drawing a reliable conclusion.

## Differential impacts on degree and non-degree holders

Figs 8 and 9 report whether changes in the five indicators differ between degree and non-degree holders. For monthly net earnings and weekly paid work hours, we analyzed an additional sample that includes only non-key workers.

As expected, the decline in earnings and paid work hours was particularly acute among non-degree holders. These differential impacts were even larger among non-key workers. When the spread of the virus decreased and most of the restrictive measures eased from July to September 2020, the difference in the impacts on non-degree and degree holders became smaller but was sustained. For paid work hours, the difference was insignificant between July and September 2020 for both the full and the non-keyworker samples. Once restrictive measures were reimposed, the difference became substantial again (p<0.001).

As Fig 9 shows, there was no significant difference in the change in subjective wellbeing between degree and non-degree holders before January 2021. However, degree holders experienced a larger increase in distress level during the third national lockdown that started in January 2021 (p = 0.028), but the differential effect disappeared in March 2021.

We do not observe a statistically significant difference in the changes in housework time between the two groups.

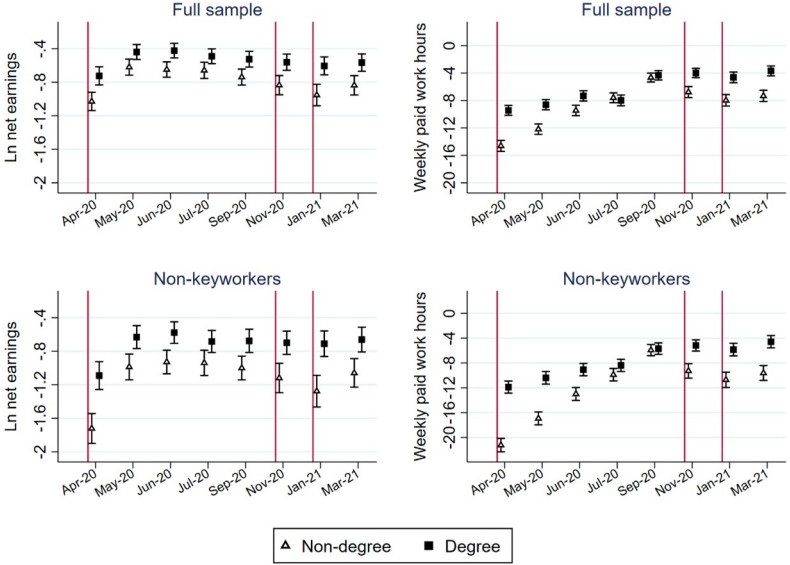

**Fig 8. Point estimates and 95% confidence interval of the education-period interaction models.**

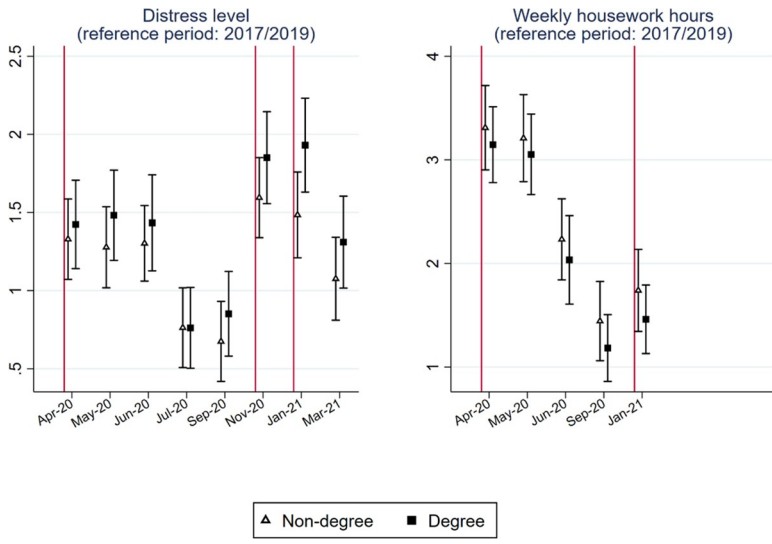

**Fig 9. Point estimates and 95% confidence interval of the education-period interaction models to predict subjective wellbeing and housework time.**

### Changes in weekly childcare hours since April 2020

Fig 10 reports whether changes in the weekly childcare hours differ across these groups.

Our findings show that women and men, BAME people and whites, and degree and non-degree holders did not differ significantly in changes to their childcare time since April 2020. However, there is a tendency that the reduction in childcare time in September, which should be associated with pupils returning to schools after summer vacation, was larger for mothers and the more educated group, suggesting that women and the more educated might have spent more time taking care of children at home.

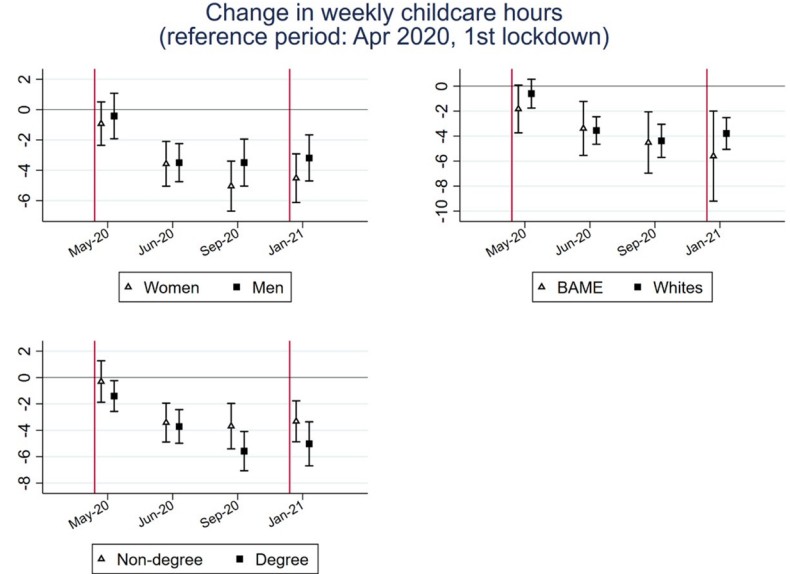

**Fig 10. Point estimates and 95% confidence interval of the gender, race, and education-period interaction models.**

For more details of the results, please refer to S2–S5 Tables. The within-individual R-squares are small when predicting earnings, subjective wellbeing, housework time, and childcare time. Small within-individual R-squares are not uncommon in fixed-effect regressions, especially when predicting housework time and subjective wellbeing [50, 51]. These results suggest that a limited number of individuals have changed their partnership and parenthood status and COVID-test results, but their outcome variables—earnings, time use, and subjective wellbeing-have changed considerably over the past year. The inclusion of more time-varying variables might be able to improve the explanatory power. Those variables could be whether furloughed, whether participated in the job retention scheme, or whether went back to work/school. However, the purpose of this paper is to provide an overall net impact of COVID-19 and its related measures on an individual instead of focusing on a specific policy or the spread of COVID-19. Given the focus on the trajectories of earnings, time use, and subjective wellbeing at different stages of the pandemic, we do not include those time-varying variables suggested above.

## Discussion and conclusion

In this article, we have utilized the latest UK COVID panel data to provide a comprehensive analysis of the dynamics of earnings, time use, and subjective wellbeing at different stages of the pandemic over the past year. Our research, with a much extended time scope, surpasses past UK studies that only followed a short period after the first lockdown imposed in March 2020 [for example, 3, 7, 9, 20]. Our analysis has incorporated multiple domains of outcomes across several social groups. We aim to examine how the spread of COVID-19 and COVID-induced policies have had unequal and dynamic impacts on different social groups in the UK. Our findings offer important insights into whether inequalities in changes in income, time use and wellbeing are likely to be long lasting or temporary.

Overall, the initial outbreak of COVID-19 and the first national lockdown brought the largest change in earnings and time use. The later two lockdowns together with the repeated new highs of the COVID-19 cases and deaths impacted people's subjective wellbeing the most. Although strict measures that aimed to reduce people's physical contact were imposed in the later two lockdowns, people's time use did not respond as strongly as they did during the first lockdown. Among the five indicators, none had returned to their prepandemic level until late March 2021. It remains uncertain when and whether earnings, working patterns, family life, and subjective wellbeing will return to the prepandemic level.

Female workers experienced less reduction in their earnings than male workers, which is largely due to the relatively high proportion of women working in key sectors, especially in the health and social care industry. Women have made an important contribution to the fight against COVID-19 by working in key sectors. However, even among non-key workers, the decline in paid work hours was smaller for women but only during the first lockdown period. These findings concur with earlier research that reported that men in the UK were more likely than women to be laid off or furloughed during the first lockdown [20]. Once lockdown measures were gradually lifted beginning in June 2020, men's paid work time recovered faster than that of women. This finding is similar to previous work on the gendered impact of natural disasters on market labor [52]. In summary, our analysis has shown that in the UK, men's paid work time was more responsive to the restrictive measures of the first lockdown, but women's and men's paid work time responded similarly in the later two lockdowns.

The subjective wellbeing of women was more sensitive to the outbreak of COVID-19 and related lockdown measures than that of men. For example, the increase in women's distress level was substantial in April, but it then gradually improved until the next lockdown. Men's responses lagged behind of those of women. Past COVID-19 research has highlighted the

gender difference in social networks, where women tend to have more friends [29]. The larger exposure to news related to COVID-19 for those with more close friends might be the factor that explains the diverging trajectories of women's and men's subjective wellbeing [53, 54]. Theses differential impacts became smaller in later two lockdowns, as the pandemic had developed for a certain period. At the beginning of the pandemic, women and men seemed to have perceived the danger of this infectious disease differently.

The gender gap in housework time was maintained over the past year. Overall, the gender-specific changes in earnings, paid work time, and subjective wellbeing were mainly observed when strict restrictions were in place, and the gender gap returned to its prepandemic level once those measures were lifted.

People of a BAME background experienced a larger loss in earnings than whites. This finding is consistent with an earlier finding on BAME immigrants in the UK [3]. We have further shown that the enlarged earning gaps between BAME and white people persisted almost over the entire year.

Persistently enlarged earning gaps were observed between non-degree and degree holders. The gap was even larger among non-key workers. Non-degree holders suffered from a larger reduction in earnings across all months over the past year. This gap was particularly large during the national lockdown periods. A similar observation was found for weekly paid work hours. The spread of COVID-19 and lockdown restrictions are associated with an enlarged gap in paid work time between non-degree and degree holders. This effect on paid work time is likely to be temporary because differential impacts were not observed from July to September 2020, when lockdown measures were mostly lifted.

One limitation of this study is that some changes could be brought by seasonal fluctuations beyond COVID-19 and its related restrictions. For example, people's paid work time in winter may differ from that in summer. General psychological health was usually worse in winter than in summer [55]. The ideal solution is to compare information collected in the same month before the pandemic and in 2020. However, this approach is not possible with the current data. If the current survey retains the current monthly or bimonthly data collection frequency, future work can compare the same month in 2020 and the years after to examine pandemic and postpandemic differences. We have also included the measure of the spread of COVID-19 (daily new cases or daily new death rates, as shown in Fig 1) to examine whether the outcomes are affected by the macrolevel development of the COVID-19 pandemic in the UK. We do not find strong evidence showing that those measures are associated with the outcomes. Our results reveal the trajectories of earnings, time use, and subjective wellbeing at different time points over the past year but cannot identify the exact impact of a specific lockdown restrictive policy. There could be other non-COVID-19-related policy updates that occurred in parallel over the past year that may have had an impact on the same outcomes. Nonetheless, the trends of the observed changes in income, time use, and subjective wellbeing corresponded closely to the different waves of the pandemic and the lockdown timeline. Therefore, the major sources of those changes should be related to the spread of COVID-19 and its related lockdown measures.

In conclusion, our findings suggest that the long-lasting pandemic and the related restrictions to contain the virus over the past year have produced persistent negative consequences for earnings, work patterns, and subjective wellbeing. The spread of COVID-19 and the national lockdowns at different stages had distinct patterns and measures, and their impacts on labor earnings, time use and subjective well-being varied. Time use patterns became less sensitive to the later lockdowns, but the distress levels reached a new high with repeated lockdowns in multiple waves of the pandemic. The differential impacts of the lockdown measures based on gender became insignificant once lockdown measures were lifted. However, some social groups, including BAME and white people and non-degree holders and degree holders,

experienced persistently enlarged gaps in earnings. The negative impacts of the spread of COVID-19 and its related measures vary not only in their extent but also in their speed among different social groups. Further research should be conducted to understand factors that have driven these social inequalities and to monitor how inequalities based on gender, educational level, and ethnic minority status might be persistent or even exacerbated in the long term.

## Supporting information

**S1 Table. Samples and sample selection.**
(DOCX)

**S2 Table. Baseline model: Changes in the five indicators across waves.**
(DOCX)

**S3 Table. Gender and period interaction model results.**
(DOCX)

**S4 Table. Ethnicity and period interaction models.**
(DOCX)

**S5 Table. Education and period interaction model results.**
(DOCX)

## Author Contributions

**Conceptualization:** Muzhi Zhou.

**Formal analysis:** Muzhi Zhou.

**Funding acquisition:** Man-Yee Kan.

**Investigation:** Muzhi Zhou.

**Methodology:** Muzhi Zhou.

**Software:** Muzhi Zhou.

**Validation:** Muzhi Zhou.

**Visualization:** Muzhi Zhou.

**Writing – original draft:** Muzhi Zhou.

**Writing – review & editing:** Man-Yee Kan.

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
