## [Decision Letter · Decision Letter 0]

4 Jan 2021

PONE-D-20-32060

The varying impacts of COVID-19 on labour earnings, time use and well-being at different stages of the United Kingdom lockdown

PLOS ONE

Dear Dr. Zhou,

Thank you for submitting your manuscript to PLOS ONE. After careful consideration, we feel that it has merit but does not fully meet PLOS ONE’s publication criteria as it currently stands. Therefore, we invite you to submit a revised version of the manuscript that addresses the points raised during the review process.

We have received recommendations of four reviewers and we ask you to incorporate these. Please also consider the documents attached, because the reviewers have directly commented in the document and supplement file.

We look forward to receiving your revised manuscript.

Kind regards,

Florian Fischer

Academic Editor

PLOS ONE

Journal Requirements:

Reviewers' comments:

Reviewer's Responses to Questions

**Comments to the Author**

1. Is the manuscript technically sound, and do the data support the conclusions?

Reviewer #1: Partly

Reviewer #2: Yes

Reviewer #3: Yes

Reviewer #4: Partly

2. Has the statistical analysis been performed appropriately and rigorously? 

Reviewer #1: Yes

Reviewer #2: Yes

Reviewer #3: Yes

Reviewer #4: Yes

3. Have the authors made all data underlying the findings in their manuscript fully available?

Reviewer #1: Yes

Reviewer #2: Yes

Reviewer #3: Yes

Reviewer #4: Yes

4. Is the manuscript presented in an intelligible fashion and written in standard English?

Reviewer #1: Yes

Reviewer #2: Yes

Reviewer #3: Yes

Reviewer #4: Yes

5. Review Comments to the Author

Reviewer #1: The purpose of this paper is to examine differential impact on societal groups of lockdown during the first wave of COVID-19 in UK. The ‘impact’ (the dependent variable of the analysis) is represented by four alternative measures of labour market participation indicators (monthly income, weekly working hours, weekly housework hours, weekly child care hours and subjective wellbeing (outcome variables). Twelve societal groups considered (independent variables) are delineated based on gender, ethnicity, educational level, and parental status. The analysis is based on a panel dataset constructed by putting together data from an earlier (pre-COVID-19) wave of the UK Household Longitudinal Survey (UKHLS) and online questionnaire survey of individuals who participated in UKHLS) conducted in March, April, and, June using the same sample frame.

The paper has examined how the impact of the lockdown varied among the identified societal groups compared to the pre-lockdown situation. This is obviously an important research issue for assessing the developmental impact of the pandemic. However, I do not find that the authors' claim that the paper provides ‘new insights into the long-term impact of the pandemic' is warranted. This is essentially a short-term analysis covering a period of just four months. The term ‘Longitudinal’ analysis is really a mismove for an analysis covering such a short period of time.

It is important to clearly spell out the dependent and independent (explanatory) variables of the analysis and the hypothesised relationship between them as part of the introductory section.

The short literature survey in the second and third paragraphs of the introduction section is not well integrated in the discussion. It is important to draw on this literaure clearly spell out the hypothesised relationships between the dependent and explanatory variables in order to set the stage for the ensuing empirical analysis. Indeed, the abasence of a clear statement of the underlying hypothesis is a major limitation of the paper.

The results reported in Figures 1 to 5 (to me these look like ‘Tables’, not ‘Figures’!).

It is not clear to me why the dependent variables are included in ‘difference’ forms, eg childless women vs childless men, fathers versus childless men, mothers versus childless men etc. Why not include each variableof societal characteristic in its own as separate variable: childless women, childless men, fathers, childless men, mothers, childless men etc. The pairwise inclusion of variables essential mean implicitly imposing the same regression coefficient on both variable in a given pair for a for a given explanatory variable, which is methodologically questionable. Including the variables separately helps appropriately interpreting the coefficients among the different societal groups based on statistical differences among them.

The results reported in the give Figures (Tables) 1 to 5 are very difficult to read because of the mmixing of OLS and Hickman results in a give colum using a very small font. You should preport the two alternative estimates in seperate colums. Also it is important to report the t-ratio or snatdad erro (or p-value) under each regression coefficent. Without this information it is not possible to meanigfully interprete the results.

Comparasition of OLS and Hickman results suggest that there are notable differences in coefficent estimates. Given that Hickman results are methodologically superior to OLS resulst (for the reason discussed in the text) why not report only the Mickman results in the text and put OLS results for comparision in an appendix.

It is important to explan how change in subjecttive wellbeing is defined/measured.

The clusion need to be rewritten adddressing the point I have made in the seond paragraph above.

Reviewer #2: This study provides a timely evaluation of changes in income, employment, household responsibilities, and general well-being throughout the course of the pandemic-related lockdown in the UK. The study advantageously uses panel data and measures monthly changes in the outcomes at each stage of the lockdown.

I believe the study provides important and interesting contributions, but the study could do more to maximize the strengths of its panel data. Otherwise, it’s not clear how it’s findings substantively improve over prior research using cross-sectional data that the study cites.

1. Clarify and Strengthen the Panel Data Analysis as the Central Contribution

a. A clearer explanation in the front end of the focus on within-person changes in the outcomes would communicate the article’s contribution more strongly. Currently, the manuscript’s front end emphasizes aggregate changes over time throughout the lockdown and inequalities across groups (pages 2-3). However, repeated cross-sectional data would be able to assess these topics. For example, repeated cross-sectional data could document “how initial inequalities by gender, ethnicity, educational level, and parental status, might have been altered at different stages of the lockdown.” Repeated cross-sections would be suitable for analyzing trends in these aggregate inequalities.

b. The panel data analyzed in this article are uniquely able to examine within-person changes between these groups—a clear advantage over cross-sectional data. But as it stands, the mean changes (within persons) the paper currently presents are not so different from changes in mean differences between groups that one could obtain from repeated cross-sectional data.

The study could strengthen the panel analysis with more detailed within-person measures. For example, measures of earnings and hours volatility within persons across several periods (like the coefficient of variation or something) would give a more complete picture of between-group differences in overall instability throughout the lockdown.

2. Heckman Selection Models

Can the authors explain the assumptions and goals of the Heckman selection models in more detail (pages 6 & 7)?

My understanding is that the selection model essentially requires variables similar to instrumental variables (affecting selection but not the outcomes) for the selection process. Otherwise, the selection model could absorb too much variation from the explanatory variables in the outcome model, undermining the outcome model estimates’ precision. It’s not clear to me whether any of the listed variables would satisfy those assumptions. Moreover, Heckman selection models can only account for observable selection processes.

Reviewer #3: This research addresses the impact in the UK of COVID-19 at various stages of the lockdown using the UK Household Longitudinal Survey (UKHLS). More specifically, these impacts are labour income, time used, and subjective well-being and are measured before, during, and after the lockdown period (i.e., before March, April, and June 2020). The authors find that that the impacts along these three metrics differ depending on individuals' gender, parental status, ethnicity and education level.

This is an important and timely piece of work which adds to the (increasingly) voluminous literature on the social impacts of COVID-19. Given its timeliness, and the limited time to think about some of the issues raised by the paper, my remarks will be brief.

My biggest concern is that the actual effects of the pandemic are not being captured in this research given the relatively short-time period under consideration. Furthermore, while causality is never assumed throughout the paper, the authors are correct in pointing out (page 3, second paragraph) that there have been many factors that have changed over this brief period “on a weekly or even daily basis . . .” That said, there is little that can be done about this given the data, short of re-estimating the models with more up-to-date data (if available). Still, I would like to see this point made clearer and that these results are not dealing with the long-term (as I understand it), despite the implication at the beginning of this paragraph that this is what the paper does. Further, if the data can be updated in a timely manner, this would be useful and give us better insights into the medium-term impacts of the lockdown.

On a related note, much of the discussion in the text has to do with the differences between the pre- and post-lockdown periods where the description is putting together the results from pre-lockdown versus April, April vs. May, and May vs. June (i.e., the three columns in Figures 1-3 and 5). I think it would be worthwhile to include a new set of results that would directly compare the pre- and post-lockdown data. These should be very similar to the results described in the text but would be much easier for the reader to interpret visually. This could easily be accomplished by adding a new column to the figures.

Minor points:

Page 2, first intro paragraph – Relative time measures should be avoided. Here it states “six months after the UK announced the lockdown on 23 March 2020 . . . “As I read this paper, it is now eight months after this date. Further, this two-month lag is an eternity given the number of papers that have been written in response to the pandemic. I would suggest that a quick update of the literature where necessary and possible.

Page 3, bottom – I assume that the measures taken in the UK were national in scope? This should be made clear since in many other countries any lockdown measures have been at the discretion of subnational levels of government.

Page 13, second paragraph – the difference in the increase in housework time of 0.6 to 0.8 hours for mothers and childless men is not statistically significant (according to Figure 3). Such results here (and elsewhere) should probably not be reported (or at the very least identified as statistically insignificant).

Page 15, paragraph 5 – do we know that childless women and mothers experienced less reductio in their working hours due to their relatively high proportions working in key sectors, or is this speculation?

Throughout – it is quite possible that there may be some seasonal effects at play in these estimates. The proper treatment of this might be to do a difference-in-difference estimation which would compare the pre- and post-lockdown period in 2020 with the same data from earlier years. I do not expect this to be done but mention of these potential effects might be worthwhile. The authors allude to these in the conclusion when they discuss needing further waves of the data.

Reviewer #4: Report on: The Varying Impacts of COVID-19 on Labor Earnings, Time Use and Well-Being at Different Stages of the United Kingdom Lockdown (MS# PONE-D-20-32060)

As the title suggests, the paper quantifies changes in well-being for a national sample during April-June 2020 when COVID-19 and related policy measures had major impacts on United Kingdom residents. Across all measures, the paper finds worse outcomes in April, a small rebound in May and little or no impact in June. The Abstract concludes “different vulnerabilities of different social groups to the lockdown measures not only affect the extent of the Covid-19 impact, but also the speed at which these effects develop.”

The paper uses good methods to document how income and behavior changed over the focal three months but is less convincing in attributing ALL the changes to the Covid-19 pandemic and subsequent lockdown measures. In other words, the paper documents correlation but not causation. A timeline of major regulatory events and their implementation lags would have been very helpful. It seems implausible that all policy measures would have instantaneous effects as currently assumed, or that individuals would not have changed their behavior voluntarily even absent government mandates. The marked-up paper identifies minor editorial suggestions. More detailed comments follow:

1. Given the stated desire to study nationwide long-term impacts of COVID-19, it is strange that the paper relies only on stale (June 2020) survey data from the United Kingdom Household Longitudinal Survey (UKHLS) rather than supplementing with more recent unemployment data from the Office of National Statistics, available monthly through November 2020. (https://www.ons.gov.uk/employmentandlabourmarket/peopleinwork/employmentandemployeetypes/bulletins/employmentintheuk/november2020)

2. Similar tests have been published for the April and May 2020 waves of the UKHLS COVID-19 surveys. The paper adds tests for the June 2020 wave and finds little changes during this month.

3. Many variables in the Supplementary are undefined making it difficult to interpret the results. The Tables would ideally be self-contained.

4. The paper should better justify why standard errors are clustered at the household level rather than by gender or region, since the effects are theorized to covary within groups.

5. The Heckman procedure for self-selection in responses seems appropriate for the research setting. However, most of the p-values from the comparison tests (8 of 14) reported in the Supplementary are well above 0.10 significance level, suggesting that the Heckman selection models have low predictive power. The paper does not discuss this low power in discussing results or drawing inferences.

6. Likewise, almost half of the OLS models (6 of 14) have model p-values that exceed the 0.10 significance level, indicating low reliability. Importantly, 4 of 5 models run on Wave 31 (July 2020) data are not significant at the 0.10 significance level. The reported R2s are often close to zero, suggesting that the adjusted R2s are negative. In other words, the reported results are like using a random number generator to measure most of the independent variables in about half the tests.

7. Interpretation of the novel June 2020 results are clouded by the problem that “absence of evidence is not evidence of absence.” Thus, while it is descriptively valid to say that there were no net changes in June 2020, this is not the same as saying that COVID-19 and the lockdown measures had no impact, because some measures such as the Coronavirus Job Retention Scheme may have fully offset the impact of COVID-19, but each of these components may have had large effects considered individually.

8. The paper should be more cautious in interpreting the results. Given that there may have been important policy changes beyond those related to COVID-19, as well as voluntary changes in behavior, all the paper reports is the net effect of all changes. The lack of change in June 2020 is still interesting, but at best the paper is still studying short-term or medium-term effects rather than long-term effects. Low explanatory power of the models should be discussed explicitly.

6. PLOS authors have the option to publish the peer review history of their article (what does this mean?). If published, this will include your full peer review and any attached files.

Reviewer #1: No

Reviewer #2: No

Reviewer #3: No

Reviewer #4: No

---

## [Author Response · Author response to Decision Letter 0]

13 Jul 2021

In this paper, we evaluate how earnings, time use, and subjective wellbeing changed from April 2020 to late March 2021, during which period the United Kingdom (UK) has experienced three COVID-19-induced lockdowns. 

We have included a new section entitled "“The impacts of the COVID-19 and its related measures” " with updated “Timeline of the lockdown measures in the UK” . In the two sections, we have reviewed latest work on the impacts of COVID-19 and its related measures, with a focus on the UK, and we have provided the development of the pandemic and policy changes in the UK over the past year.

We have updated our data to cover the time point from before the pandemic up to late March 2021, therefore coving the entire first year of the pandemic.

We have now employed a simpler fixed-effect regression approach to document within-individual changes in each time point over the past year. We noted that the Heckman models did not add much to our findings and thus dropped it.

Accordingly, the figures and tables have now been substantially revised.

For a more detailed point-by-point respose, please refer to the uploaded response letter.

---

## [Decision Letter · Decision Letter 1]

2 Aug 2021

PONE-D-20-32060R1

The varying impacts of the three UK COVID-19 lockdowns: A year in review

PLOS ONE

Dear Dr. Zhou,

Thank you for submitting your manuscript to PLOS ONE. After careful consideration, we feel that it has merit but does not fully meet PLOS ONE’s publication criteria as it currently stands. Therefore, we invite you to submit a revised version of the manuscript that addresses the points raised during the review process.

Please find attached additional comments raised by Reviewer 4 in the documents attached. Incorporating these comments will help to improve the quality of your paper.

We look forward to receiving your revised manuscript.

Kind regards,

Florian Fischer

Academic Editor

PLOS ONE

Journal Requirements:

Reviewers' comments:

Reviewer's Responses to Questions

**Comments to the Author**

1. If the authors have adequately addressed your comments raised in a previous round of review and you feel that this manuscript is now acceptable for publication, you may indicate that here to bypass the “Comments to the Author” section, enter your conflict of interest statement in the “Confidential to Editor” section, and submit your "Accept" recommendation.

Reviewer #3: All comments have been addressed

Reviewer #4: (No Response)

2. Is the manuscript technically sound, and do the data support the conclusions?

Reviewer #3: Yes

Reviewer #4: Yes

3. Has the statistical analysis been performed appropriately and rigorously? 

Reviewer #3: Yes

Reviewer #4: I Don't Know

4. Have the authors made all data underlying the findings in their manuscript fully available?

Reviewer #3: Yes

Reviewer #4: Yes

5. Is the manuscript presented in an intelligible fashion and written in standard English?

Reviewer #3: Yes

Reviewer #4: No

6. Review Comments to the Author

Reviewer #3: All comments and suggestions from my original review have been completely to my satisfaction. This is an interesting piece and should be well-received.

Reviewer #4: (No Response)

7. PLOS authors have the option to publish the peer review history of their article (what does this mean?). If published, this will include your full peer review and any attached files.

Reviewer #3: No

Reviewer #4: No

---

## [Author Response · Author response to Decision Letter 1]

20 Aug 2021

We have revised the manuscipt to incorporate Reviewer 4's comments. We have revised some wordings and added more sentences to highlight that not ALL changes observed in the results are simply due to lockdown policies. We would like to emphasize that we aim to document the trends of these outcomes at different time points over the past year to shed light on the longer-term changes in those outcomes. This is an advancement from previous studies. We do not try to argue that we provide an exact estimation of these impact. We also have no intention to separate the impact of the spread of COVID-19 in the UK and policies that aim to contain this virus. In line with most of the COVID-studies cited in this paper, we analyzed outcomes at different time points over a period to improve our understanding of the impact of the pandemic on people’s lives. We believe that this analysis has offered important foundation to understand the potential impacts of this pandemic.

---

## [Editor Report · Decision Letter 2]

31 Aug 2021

The varying impacts of COVID-19 and its related measures in the UK: A year in review

PONE-D-20-32060R2

Dear Dr. Zhou,

We’re pleased to inform you that your manuscript has been judged scientifically suitable for publication and will be formally accepted for publication once it meets all outstanding technical requirements.

Kind regards,

Florian Fischer

Academic Editor

PLOS ONE
---

## [Editor Report · Acceptance letter]

3 Sep 2021

PONE-D-20-32060R2 

The varying impacts of COVID-19 and its related measures in the UK: A year in review 

Dear Dr. Zhou:

I'm pleased to inform you that your manuscript has been deemed suitable for publication in PLOS ONE. Congratulations! Your manuscript is now with our production department. 

Kind regards, 

on behalf of

Dr. Florian Fischer 

Academic Editor

PLOS ONE